

# A conditional random field based approach for high-accuracy part-of-speech tagging using language-independent features

Mushtaq Ali[1], Muzammil Khan[1] and Yasser Alharbi[2]

[1] Department of Computer and Software Technology, University of Swat, Swat, KP, Pakistan
[2] College of Computer Science and Engineering, University of Hail, Hail, Saudi Arabia

## ABSTRACT

Part-of-speech (POS) tagging is the process of assigning tags or labels to each word of a text based on the grammatical category. It provides the ability to understand the grammatical structure of a text and plays an important role in many natural language processing tasks like syntax understanding, semantic analysis, text processing, information retrieval, machine translation, and named entity recognition. The POS tagging involves sequential nature, context dependency, and labeling of each word. Therefore it is a sequence labeling task. The challenges faced in Urdu text processing including resource scarcity, morphological richness, free word order, absence of capitalization, agglutinative nature, spelling variations, and multipurpose usage of words raise the demand for the development of machine learning automatic POS tagging systems for Urdu. Therefore, a conditional random field (CRF) based supervised POS classifier has been developed for 33 different Urdu POS categories using the language-independent features of Urdu text for the Urdu news dataset MM-POST containing 119,276 tokens of seven different domains including Entertainment, Finance, General, Health, Politics, Science and Sports. An analysis of the proposed approach is presented, proving it superior to other Urdu POS tagging research for using a simpler strategy by employing fewer word-level features as context windows together with the word length. The effective utilization of these features for the POS tagging of Urdu text resulted in the state-of-the-art performance of the CRF model, achieving an overall classification accuracy of 96.1%.

## INTRODUCTION

Part-of-speech (POS) tagging is the process of assigning tags or labels to each word of a sentence based on its grammatical category. These labels can be used for associating each word to its corresponding grammatical category (*i.e.*, POS) in a given text (*Warjri et al., 2021*). POS tagging is an essential and often a prerequisite step in many natural language processing (NLP) applications including text analysis, syntax understanding, semantic analysis, information retrieval, machine translation and named entity recognition.

 Due to its pivotal role in many NLP tasks, much attention has been given in the recent past to achieving accurate and efficient POS tagging in many different languages of the world. POS tagging has been widely covered and much advancement achieved for most

Corresponding author
Muzammil Khan,
muzammilkhan86@gmail.com

Western languages. However, for resource-scarce languages like Urdu little work has been done particularly in the application of contemporary machine learning (ML) and deep learning approaches for Urdu POS tagging.

Urdu is the national language of Pakistan and is widely spoken across the world. POS tagging is a sequence labeling task as it involves a sequential nature, context dependency, and labeling of each word within the text. Many different approaches including supervised learning, unsupervised learning, hybrid of rule-based and machine learning, and deep learning based approaches have been followed for Urdu POS tagging. Supervised Learning techniques require labeled data for training. Among the traditional Supervised techniques, HMM, maximum entropy models (MaxEnt), and support vector machine (SVM) are popularly used for Urdu POS tagging. In this work, an Urdu POS classifier also called tagger is developed for classification and prediction of POS tags of Urdu news text using the Mushtaq & Muzammil POS Tagged (MM-POST) dataset (*Ali & Khan, 2024a*) available at *Ali & Khan (2024b)*. The simple language-independent and smaller feature set has been selected for the training of the CRF model to learn the pattern and predict the Urdu POS tags from the real Urdu text. The CRF is a simple probabilistic graphical model popular for segmentation and labelling tasks due to its less computation and low requirement of extensive feature engineering.

## Challenges of Urdu POS tagging

The generally faced challenges that make the POS tagging of the low-resource Urdu language a difficult task include resource scarcity, morphological richness, no capitalization, free word order, agglutinative nature, spelling variations, and words serving multiple grammatical functions (*Malik & Sarwar, 2015*; *Shah et al., 2016*). Due to the complex morphological and syntactic structure of Urdu, POS tagging requires careful handling due to the ambiguity in word categorization. The scarcity of large-quality annotated corpora, need for use of sophisticated features set and the requirement of huge processing power for running heavy computations in learning and predicting the POS tags make Urdu POS tagging a challenging task. Conditional random fields (CRF), a sophisticated supervised machine learning algorithm, is widely used for POS tagging in Western languages and has also been successfully applied to Urdu POS tagging. A CRF-based POS classifier has been developed for 33 Urdu POS categories, utilizing language-independent features and trained on the MM-POST Urdu news dataset, demonstrating its effectiveness in handling Urdu's linguistic complexities.

## Motivation for study

POS tag provides linguistic information about how a word can be used in a phrase, sentence or document. It helps in identifying grammatical context of the text that is highly effective in text prediction and generation. POS tagging is an essential part of many state-of-the-art NLP applications like text analysis, syntax understanding, semantic analysis, information retrieval, machine translation, text to speech systems, question answering, sentiment analysis and named entity recognition.

In the past, various rule-based approaches have resulted in encouraging performance for Urdu POS tagging. However, rule-based systems are difficult to develop and are less portable to other domains. Notable performance has been reported by various researchers using ML and deep learning approaches for Urdu POS tagging but limited application of various ML and deep learning techniques is found for Urdu POS tagging due to the existing challenges. Therefore, exploring the area of Urdu POS tagging can lead to feasible, efficient and automatic solutions.

The tagset of 12 Urdu POS categories having 32 subcategories and POS tagging of 100,000 words of Urdu Digest Corpus made through the Tree Tagger using decision tree and the Center for Language Engineering (CLE) tagset, achieving an accuracy of 96.8% (*Ahmad et al., 2014*). In our work, these main categories and subcategories of the CLE Tagset have been used for annotation of the training dataset and the effectiveness of these POS categories is utilized for learning and prediction of labels through ML based Urdu POS tagging system.

Different lexical word level, lexical character level, ngram and word-embeddings have been used as features in the literature for Urdu POS tagging. *Adeeba, Hussain & Akram (2016)* inferred that lexical features are more effective than structural features, and that the size of training data affects the accuracy as larger data improves the performance accuracy. In our previous work, an Urdu POS tagged dataset, the Mushtaq & Muzammil POS Tagged (MM-POST) dataset, comprising of 119,276 words or tokens has been developed (*Ali & Khan, 2024a*). The annotated data and a good feature set are the key requirements for any machine learning classification system (*Khan et al., 2019a*). Therefore, an effective simple, smaller, language independent word level features-set has been selected for devising a ML classification system to automatically identify and predict POS tags of Urdu news text using the MM-POST dataset.

## Motivation for choice of method used

The rule-based, machine learning based, and deep learning-based approaches have been used by the researchers for POS tagging of the post-positional and morphologically rich Urdu language. Modern machine learning and deep learning techniques are useful due to their portability across domains. The supervised ML approaches require large, labelled data for training models and automatically inducing rules in a shorter time than rule-based and deep learning approaches. The use of the CRF as one of the more advanced Supervised ML algorithms is widely found for POS tagging of western languages. Few researchers have also effectively demonstrated the use of CRF technique for POS tagging of Urdu text. Utilization of the correlativity inside two POS tags and capturing of sequential dependencies in a less computational time make CRF a suitable choice for POS tagging.

The CRF has been preferred over other ML techniques and has been reported by the research community of Urdu POS tagging to achieve state-of-the-art performance. Therefore, CRF technique has been selected for implementation of our Urdu POS tagging system. CRF becomes the better choice in comparison to Neural Networks, Transformers or other deep learning based models for tasks like Urdu POS tagging that involves training data scarcity, availability of less computational resources, sequence labelling and

handling rich morphology among other challenges. However, if these challenges of Urdu language are handled, the neural networks, Transformers and other deep learning models can offer many potential advantages like automatic feature extraction, contextualized embeddings, handling long range dependencies, providing transfer learning and better generalization *etc.*

The lexical features have been reported as more effective than structural features. The lexical word level features are easy to determine and does not require linguistic knowledge or heavy computation. In the literature, language dependent, language independent and mixed types of features sets have been used. The structural and language-dependent features are more complex, many in number and large sized. They are difficult to determine and require huge computational resources. In contrast, in our work, simple, fewer and easy to find word level language-independent features have been used. The feature set includes context word window features along with the Word Length feature for each token for training and testing of a supervised machine learning model with a moderate sized Mushtaq & Muzammil POS Tagged (MM-POST) dataset (*Ali & Khan, 2024a*). The 33 POS categories of the CLE tagset have been used as labels.

The selected smaller features-set of five language-independent features has been used to train and test the CRF model for Urdu POS tagging using the MM-POST annotated dataset. The features set comprising of one immediately preceding lexical token (*i.e.*, previous lexical token) and two successive next lexical tokens (*i.e.*, next and second next tokens) of the current word or token have been utilized to serve as the context window together with the Word Length of the current word or token have been used for learning and prediction of the POS tag for every current word or token of the dataset.

Our work is different from other researchers in terms of the number and complexity of features used to train the machine learning models. We used simple and fewer features (five in number) for the training and testing of the model. These features are language independent *i.e.*, the features selection and understanding do not require linguistic knowledge. It provides a wider scope for the use of the selected features set for experimentation with many different tasks and techniques. Similarly, less number of features are used to learn the pattern of the Urdu POS tags and ensure their prediction in an optimal computation time. This allows the expansion of the application of the pursued approach for much larger datasets in the future. The Urdu news dataset, the MM-POST, has been used for training and testing of the model because the news text is formal and is rich in occurrences of different POS tags as compared to other genres. The informal text has inconsistent syntax, less accuracy of words and much noisy data. In contrast, the news text has consistency in words structure, less noisy data and richness of POS occurrence that make the news text a preferred choice in achieving better model performance in POS tagging of scarce resourced Urdu language text. To effectively tackle POS tagging for informal text, targeted experiments must adapt techniques from formal text to suit the distinct characteristics of informal language. This involves refining preprocessing methods and feature extraction strategies to improve the accuracy and robustness of models applied to informal datasets.

The 33 different grammatical categories or POS tags selected from the POS tagset of the Center for Language Engineering (CLE), in the creation and labeling of the MM-POST dataset, have been used as POS labels for the model's training and prediction. Instead of relying on the requirements of large data, computational resources and sophisticated techniques that are difficult to interpret, we have built an efficient Urdu POS classifier that can effectively predict the POS label of Urdu text for the training dataset as well as the unseen validation data. Our approach benefits from the use of simple language independent features like context word window and word length utilizing medium sized dataset making the proposed approach extendable to other resource-scarce languages.

The article is organized as follows. 'Related Work' provides a survey of the related work regarding Urdu POS tagging. 'Research Methodology' discusses the research methodology adopted for this work and the evaluation of results is presented in 'Evaluation of Results'. Finally, the 'Conclusion and Future Work' section concludes the article and provides directions for future work.

## RELATED WORK

In the literature, the research approaches pursued for Urdu POS tagging include rule-based, machine learning-based, and hybrid approaches.

### Rule-based approach

In rule-based approach manual contextual rules are created for tagging words using their lexical information. The rule-based approach provides the advantages of explainability, customizability and no need of training data. However, Rule-based Urdu POS tagging requires linguistic knowledge, expertise in rule synthesizing, and a long development time. The rule-based approach is good for domain-specific work but the tagged results are less portable to other domains (*Khan et al., 2019b*). The labor intensiveness, limited scalability, and difficulty in hand-crafting exceptions and irregularities occurring in Urdu language make machine learning based approach a better alternative to the rule-based approach.

### Machine learning based approach

Modern machine learning and deep learning techniques are useful due to their portability across domains. However, limited application of various machine learning techniques is found for Urdu POS tagging due to the scarcity of large-quality annotated corpora. Cross-validation through bootstrapping of the manually tagged data with the automatic tagging can be used to leverage the lack of annotated data for a less-resourced language like Urdu (*Baig et al., 2020*). Cross-validation through bootstrapping is beneficial for low resourced languages like Urdu for its resource efficient validation, facilitated error analysis, reduced Overfitting and effective maximizing of limited data. The first ever ML work on Urdu POS by *Anwar et al. (2007)* proposed a statistical approach based on the n-gram Markov model, using the Enabling Minority Language Engineering (EMILLE) *corpus* for training and testing and two separate tagset used, comprising 250 and 90 tags each, achieved best accuracy of 95%. The large tagset of 250 tags had morpho-syntactic features whereas the smaller tagset with 90 tags was reconstructed from the former by including only the basic

 

POS, eliminating the least occurring tags and modifying and combining some tags. The purpose of using the reconstructed smaller tagset was to reduce the information for processing and improvement of model performance. Using the large Tagset, the accuracy of 91%, 83% and 91.6% was achieved for Unigram, Bigram and Backoff models respectively. However, for smaller tagset the accuracy for Unigram, Bigram and Backoff was reported better as 94.3%, 88.5% and 95% respectively. Comparing the performance of four taggers including Tree tagger, Random Forest (RF) tagger, TnT tagger, and SVM-4 tagger by *Sajjad & Schmid (2009)* using a *corpus* of 110,000 web items through 42 tags, reported SVM tagger to be the best among all with an accuracy of 95.66%. SVM outperformed other tagging methods owing to its capacity to identify differences at the phrase level within the text. By considering not only the neighbouring tags but also the surrounding words, SVM effectively captured contextual relationships, leading to enhanced tagging accuracy and overall superior performance. *Muaz, Ali & Hussain (2009)* developed a new Urdu Tagset and a *corpus* of 230,000 words by combining two corpora using the Tnt and Tree POS taggers for POS tagging and reported an accuracy of 94.2% on their new Tagset for individual corpora and 91% for the combined *corpus*. *Jawaid, Kamran & Bojar (2014)* extended the work of *Jawaid & Bojar (2012)* by performing automatic Urdu POS tagging using SVM on the text of 5.4 million sentences with 95.4 million words crawled from BBC Urdu, Urdu Planet, and other sites. They proposed a standalone POS tagger achieving a POS tagging accuracy of 88.74%. *Jawaid & Bojar (2012)* used ensemble of three taggers Shallow Parser (termed as SH Parser) developed by Language Technologies Research Centre of IIIT Hyderabad, the HUM Analyzer and the SVM. The final tag was obtained as a result of voting among the results of the three taggers. They used CRULP data of 123,843 tokens for Training and (*Sajjad & Schmid, 2009*) data comprising of 8,670 tokens for testing. Tagging every token by all three taggers and voting among the results seem impractical. Therefore, the extension of the work of *Jawaid & Bojar (2012)* in *Jawaid, Kamran & Bojar (2014)* includes release of a sizeable *corpus*, consolidation of the tagging result of the three taggers to form a standalone tagger and performing training and testing of the SVM model using the standalone tagger and the large sized *corpus*. The performance of different Urdu POS taggers heavily depends upon the Tagset used, the size and structure of the *corpus* utilized for training and testing and the model chosen. *Adeeba, Hussain & Akram (2016)* performed automatic genre identification for culture, science, religion, press, health, sports, letters, and interviews of Urdu documents through analysis of lexical (words unigram & bigram and TFIDF) and structural (words POS & sense) features. They applied SVM, Naive Bayes, and C4.5 on two datasets, *i.e.*, the CLE Urdu digest 100k words and the CLE Urdu digest 1 million words. They concluded that SVM outperforms other classifiers irrespective of feature type, the lexical features are more effective than structural features, and that the size of training data affects the accuracy as larger data improves the performance accuracy. For SVM they reported the F-measure as 0.70.

The annotated data and a good feature set are the key requirements for any machine learning classification system (*Khan et al., 2019a*). The performance of machine-learning/ statistical models for POS tagging mainly depends on the domain of the training set, the

Tag set used for annotation, and the size of the dataset (*Daud, Khan & Che, 2017*; *Mukund, 2012*; *Khan et al., 2016*).

The POS taggers have better performance on structured and well-edited data than on unstructured data. *Baig et al. (2020)* conducted a comparison of the performance of the two taggers, the IIT Urdu Shallow tagger and the CLE Statistical POS tagger, on news text and tweets data. They reported higher accuracy for both the Taggers in the case of well-edited news text than the tweets as shown in Table 1.

Structured data like news data lead to better performance of tagger because it has consistent patterns of words and sentences that provide clues to the tagger in understanding the structure of words, the linguistic patterns and grammatical rules. The unstructured data like social media posts, text messages/sms and personal blogs have informal and less organized structure of data. The taggers have low performance due to the challenges faced in handling of unstructured data like ambiguity, lack of context and noise in the data. The unstructured data require explicit sophisticated techniques for handling of these challenges.

This emphasizes the significance of the availability of structured Urdu data in different domains for better POS tagging results.

## Hybrid approach

*Naz et al. (2012)* pioneered the use of transformation-based learning (TBL) for Urdu POS tagging by employing the TBL algorithm for the automatic generation of rules from training data. The TBL is a non-probabilistic local decision system using both rules and statistical models. They used a rule-based approach and statistical models as a hybrid for the automatic generation of rules with training data of 123,755 words using 36 tags and achieved an accuracy of 84%. The strengths of TBL include its effectiveness for small datasets, incremental learning, easy to interpret and robustness to noise. However, the weaknesses of TBL including dependence on initial tags, scalability issues, limited contextual awareness and less effectiveness for highly variable data need to be regarded. *Jawaid & Bojar (2012)* used the linguistic rule-based approach together with SVM with a voting scheme for Urdu POS tagging for the tagged data from the Center for Research in Urdu Language Processing (CRULP). They compared their approach with a morphological analyzer and Urdu parser and reported an accuracy of 87.98% for their work.

The Center for Language Engineering (CLE) tagset (*Center for Language Engineering, 2023*) created by improving the versions from *Sajjad & Schmid (2009)* and *Muaz, Ali & Hussain (2009)* has 12 main syntactic categories of noun, pronoun, nominal modifiers, verb, auxiliaries, adposition, adverb, conjunction, interjection, particle, symbol and residual. These main categories are further divided into 35 subcategories as listed in Table 2. The Urdu tagsets earlier than CLE tagset were mainly adapted form other English *corpus* like Brown Corpus Tagset and Penn Treebank tagset. They lacked linguistic categories needed to handle the complex morphology and syntactic structure of Urdu. The CRULP Urdu POS tagset was one of the first attempts to develop a specific tagger for Urdu language, but it included limited in depth categories. The CLE Tagset specifically designed for Urdu language has more acceptance due to its standard set of categories,

**Table 1 Performance of IIT Urdu shallow tagger and CLE POS tagger.**

| Tagger | Evaluation metrics | News text | Urdu tweets |
|---|---|---|---|
| IIIT Urdu shallow tagger | Precision | 95.4% | 66.6% |
| | Recall | 96.7% | 64.7% |
| | F-measure | 96.1% | 65.6% |
| CLE statistical POS tagger | Precision | 93.4% | 60.6% |
| | Recall | 94.6% | 62.2% |
| | F-measure | 94% | 61.5% |

comprehensive coverage and detailed linguistic phenomena that make the CLE tagset suitable for many NLP tasks. The proposed research work benefits from the utility of the CLE tagset.

A new tagset of 12 Urdu POS categories designed with 32 subcategories and POS tagging of 100,000 words of Urdu Digest *Corpus* is made through the Tree Tagger using the decision tree and the CLE tagset, achieving an accuracy of 96.8% (*Ahmad et al., 2014*).

A Supervised POS tagger for Urdu social media content has been developed by *Baig et al. (2020)* with a focus on POS tagging of Urdu tweets, introducing a new Tag set for POS tagging of Urdu tweets and creating a tagged *corpus* of 500 Tweets from the domains of business, entertainment, politics, and sports, *etc.* They used bootstrapping in addition to manual tagging, to overcome the shortage of annotated data. The Stanford POS tagger is used for tagging of the Urdu Tweets, reporting 93.8% precision, 92.9% recall, and 93.3% F-measure.

The first conditional random field (CRF) based approach proposed by *Khan et al. (2019b)* for Urdu POS tagging used two types of features: language-dependent or linguistic features (*i.e.*, POS tag of the previous word and suffix of the current word) and language-independent feature (*i.e.*, context words window). They used ten unigram templates for feature set generation. Their features set included "Previous Lexical Word", "Current Lexical Word", "Next Lexical Word", "Current Lexical Word + Previous Lexical Word", "Current Lexical Word + Next Lexical Word", "Current Lexical Word + N-1 and N-2 Previous Words", "Current Lexical Word + N+1 and N+2 Next Words", "Part of Speech tag of Previous Lexical Word", "Suffix of Current Lexical Word" and "Length of Current Lexical Word". They termed the morpho-syntactic ambiguity or dual behavior of Urdu POS tags as the major challenge. The two datasets used in their work are the CLE dataset and the Bushra Jawaid (BJ) dataset and evaluated the performance of the CRF technique against the baseline SVM of *Jawaid, Kamran & Bojar (2014)* for Urdu POS and reported an accuracy of 88.74% with an improvement of 8.3% to 8.5% over the F-measure of the baseline SVM. Developing a strong feature set enhances the highest level of intelligence and a good feature set is more important that model itself. *Khan et al. (2019b)* proposed a balanced feature set of both the language dependent and language independent features and demonstrated the impact of the selected features set on the performance of the CRF

**Table 2 The CLE Tagset.**

| Categories | Types | POS Tag |
|---|---|---|
| 1. Noun | 1.1 Common | NN |
| | 1.2 Proper | NNP |
| 2. Verb | 2.1 Main verb infinitive | VBI |
| | 2.2 Main verb finite | VBF |
| 3. Auxiliary | 3.1 Aspectual | AUXA |
| | 3.2 Progressive | AUXP |
| | 3.3 Tense | AUXT |
| | 3.4 Modals | AUXM |
| 4. Pronoun | 4.1 Personal | PRP |
| | 4.2 Demonstrative | PDM |
| | 4.3 Possessive | PRS |
| | 4.4 Relative demonstrative | PRD |
| | 4.5 Relative personal | PRR |
| | 4.6 Reflexive | PRF |
| | 4.7 Relative apna | APNA |
| 5. Nominal modifier | 5.1 Adjective | JJ |
| | 5.2 Quantifier | Q |
| | 5.3 Cardinal | CD |
| | 5.4 Ordinal | OD |
| | 5.5 Fraction | FR |
| | 5.6 Multiplicative | QM |
| 6. Adverb | 6.1 Common | RB |
| | 6.2 Negation | NEG |
| 7. Adposition | 7.1 Preposition | PRE |
| | 7.2 Postposition | PSP |
| 8. Conjunction | 8.1 Coordinate conjunction | CC |
| | 8.2 Subordinate conjunction | SC |
| | 8.3 SCKar | SCK |
| | 8.4 Pre-sentence | SCP |
| 9. Interjection | 9.1 Interjection | INJ |
| 10. Particle | 10.1 Common | PRT |
| | 10.2 Vala | VALA |
| 11. Symbol | 11.1 Common | SYM |
| | 11.2 Punctuation | PU |
| 12. Residual | 12.1 Foreign fragment | FF |

model has been demonstrated. They compared the performance of their CRF approach with the baseline SVM and concluded that CRF outperformed the SVM.

In their work *Khan et al. (2019a)* provided a comparison of machine learning and deep learning approaches for Urdu POS tagging, using word embeddings and the context word window as features for CRF and DRNN models on two Urdu datasets the CLE dataset and

Bushra Jawaid dataset. Their eight context word features included: (1) the token (the current word), (2) the word to the left of the current word, (3) the word to the right of the current word, (4) Joint use of the Current word and the word to the left of the current word, (5) Joint use of the current word and the word to the right of the current word, (6) Joint use of the current word and N-1, N-2 left words of the current word and (7) Joint use of the current word and N+1, N+2 right words of the current word. They inferred that on the CLE dataset, the CRF performed better than the SVM, RNN, and n-gram approaches whereas the DRNN had better results on the Bushra Jawaid dataset. They argued that the utilization of correlativity inside two tags by CRF enabled it to perform better than SVM and RNNs. On the other side, SVM utilizes the maximal margin conception to have the capacity to manage the whole observation at a time. They reported that for the CLE dataset, the CRF gave a better accuracy of 83.52% than the averaged accuracy achieved by SVM, LSTM-RNN, LSTM-RNN with CRF output and HMM models of 78.12%, 75.64%, 75.06% and 75.03% respectively. However, on the BJ dataset, the LSTM-RNN resulted in a better average accuracy of 88.7% than SVM, RNN variants, CRF, and HMM average accuracy of 83.75%, 88.09%, 88.4% and 88.19% respectively.

*Nasim, Abidi & Haider (2020)* proposed Urdu POS taggers for the two models *i.e.*, CRF and BiLSTM with CRF on the Bushra Jawaid (BJ) dataset having 5.4 million sentences with 610,275 unique words, using 40 POS tags. They utilized the feature set including Word, Length, Is_Firest, Is_Last, Suffix, Prev_Word_1, Prev_Word_2 and Next_Word of the current word and reported an F1-score of 96% for both of the models, claiming their BiLSTM-CRF approach surpassing accuracy achieved for SVM (88.74%) by *Jawaid, Kamran & Bojar (2014)* and CRF (93.56%) by *Khan et al. (2019b)*. However, the accuracy achieved for BiLSTM-CRF (96.3%) was slightly better than for their CRF model (95.8%).

The accuracy reported for decision tree is 96.8% using CLE Urdu digest *corpus* with smoothing technique of class equivalence for Urdu POS tagging through their new designed Tagset (*Ahmad et al., 2014*). The better performance accuracy of CRF has been 95.8% using BJ dataset (*Nasim, Abidi & Haider, 2020*) and accuracy of 88.7% is reported for CRF using CLE POS tagged dataset and BJ dataset (*Khan et al., 2019b*). The SVM model achieved an accuracy of 95.6% (*Sajjad & Schmid, 2009*) for 110,000 tokens taken from a news *corpus* (https://jang.com.pk/). The decision tree and SVM models involve more computational complexity than CRF and require complex features engineering together with large training *corpus*. However, CRF is efficient due to its characteristics of sequence modelling and is less complex due to probabilistic graphical modelling of dependencies. CRF has proved effective in Urdu POS tagging for moderately large datasets. The RNN achieved better accuracy of 88.1% for BJ dataset (*Khan et al., 2019a*) confirming the requirement of larger datasets for application of the Deep Learning techniques. Table 3 provides a summary of the performance achieved through different techniques employed in the research community for Urdu POS tagging using different datasets that reveals decision tree, BiLSTM+CRF, CRF, SVM, and n-gram Markov have been performing better among other machine learning and deep learning techniques.

Different approaches including rule-based, ML and hybrid of the two have been used for Urdu POS tagging. The morphological and structural challenges of Urdu language require

**Table 3  Urdu POS tagging techniques and results.**

| Researcher | Corpus | Technique | Accuracy | F1-score |
|---|---|---|---|---|
| Khan et al. (2019b) | | CRF | 88.74% | 8.3% to 8.5% improved from SVM |
| Baig et al. (2020) | Tweets corpus | Stanford tagger | | 93.3% |
| Anwar et al. (2007) | EMILLE | n-gram Markov | 95% | |
| Naz et al. (2012) | | TBL | 84% | |
| Sajjad & Schmid (2009) | | SVM | 95.66% | |
| Muaz, Ali & Hussain (2009) | | TnT & Tree taggers | 94.2% | |
| | | Combined corpus | | |
| Jawaid & Bojar (2012) | | SVM plus rule based | 87.98% | |
| Jawaid, Kamran & Bojar (2014) | Bushra Jawaid | SVM | 88.74% | |
| Khan et al. (2019a) | CLE dataset | CRF | 83.52% | |
| | | SVM | 78.12% | |
| | | LSTM-RNN | 75.64% | |
| | | LSTM-RNN+CRF | 75.06% | |
| | | HMM | 75.03% | |
| Khan et al. (2019a) | Bushra Jawaid | LSTM-RNN | 88.77% | |
| | | SVM | 83.75% | |
| | | RNN | 88.09% | |
| | | CRF | 88.4% | |
| | | HMM | 88.19% | |
| Ahmad et al. (2014) | CLE Urdu digest | Decision tree, tree tagger | 96.8% | |
| Nasim, Abidi & Haider (2020) | Bushra Jawaid | BiLSTM+CRF | 96.3% | 96% |
| | | CRF | 95.8% | 96% |

the availability of sufficiently large, labelled dataset for training, testing and evaluation of supervised and other learning techniques for processing of POS tags. Many researchers adopted the tagsets and techniques of other western languages for application in Urdu POS tagging. They contributed to opening doors for further research by laying the foundation based on approaches of other languages. Different learning models including SVM, hidden Markov model (HMM), decision tree, conditional random field (CRF), recurrent neural network (RNN), long short-term memory (LSTM) among others have been used in the past for Urdu POS tagging. However, the specific challenges of Urdu languages need linguistic resources and sophisticated techniques and tools. Limited application of modern ML or deep learning methods have been witnessed for POS tagging of the resource scarce Urdu language. The use of a standard, suitable and comprehensive Tagset has been one of the challenging limitations together with selection of an appropriate language independent or language dependent features set for Urdu language in the research community. The Urdu POS tagging has high potential in improving accuracy, computational efficiency, covering the structured and unstructured domains, standardizing the tagset, building quality corpora and devising new tools and frameworks. The supervised machine learning approaches make use of large pre-labeled data for training models to learn patterns and

automatically induce rules within a shorter time than the rule-based approach. However, low-resource languages like Urdu lag far behind in the provision of large labeled quality data or corpora. Therefore, an Urdu POS tags classifier is built using a supervised CRF-based technique.

# RESEARCH METHODOLOGY

The research methodology for Urdu POS tagging through a supervised CRF-based learning approach involves the use of a dataset for training, testing, and evaluation of the model, the selection of a features-set, and experimentation for evaluation of the proposed approach as explained below.

## Urdu POS tagging through CRF

The conditional random fields (CRF) is a probabilistic graphical model suitable for segmentation and sequence labeling tasks. The CRF is characterized by its simplicity for its less computation time and low requirement for extensive featuring engineering, thereby minimizing the workload of human experts (*Khan et al., 2019a*). The CRF is an advanced supervised ML algorithm that can capture sequential dependencies among data points and is used for Urdu POS tagging as one of the more advanced techniques.

When utilizing CRF for POS tagging, the tokens are represented as an observation sequence:

X = (x1, x2, . . . , xn) and labeled as tag sequence Y = (y1, y2, . . . ., yn), CRF model aims to identify the label y that maximizes the Conditional Probability of Y given X, for the sequence X and is mathematically expressed (*Khan et al., 2019b*) as shown in Eq. (1).

$$P(Y|X) = \frac{1}{Z(X)} \exp\left( \sum_{i=1}^{n} \sum_{j=1}^{m} \lambda_j f_j(y_i, y_{i-1}, X, i) \right) \quad (1)$$

where:

$P(Y|X)$ is the conditional probability of the output sequence $Y$ given the input sequence $X$.

$Z(X)$ is the normalization factor or partition function.

$\lambda_j$ represents the parameters or weights associated with feature functions $f_j$.

$f_j$ are feature functions capturing dependencies between neighboring variables in the sequence.

The application of the CRF model has been demonstrated for the Urdu part-of-speech tagging in the research community.

In this research work, the CRF model has been trained and tested on the MM-POST (Mushtaq & Muzammil POS Tagged) dataset, using the word level language-independent features of the current word/token as context window together with the Word Length of the current word.

## The dataset

The MM-POST dataset has been used for training and testing of the CRF model for Urdu POS tagging. The dataset contains POS-labeled data from seven different news domains of

the Urdu language including Entertainment, Finance, General, Health, Politics, Science, and Sports with 119,276 total tokens for 2,871 sentences (*Ali & Khan, 2024a*) as shown in Table 4. The number and percentage shares of POS tags of different news domains in the MM-POST dataset are graphically shown in Fig. 1.

The tokenization has been already done in our previously developed dataset, the MM-POST dataset and the tokenized lexical words with their corresponding POS tags are readily available for use. The tokenization of well-structured news data resulted into well edited, consistent and useful tokens to be used for training and testing of machine learning models for any of the sophisticated Urdu NLP tasks. Our proposed CRF model for Urdu POS tagging was trained and tested using the tokenized data of the MM-POST. The necessary preprocessing for normalization of the dataset has been already done by removing extra spaces and unnecessary characters from individual words and manually correcting the inconsistent or incorrect words in the dataset. Thus, relieving the need for separate tokenization and pre-processing of the data. The dataset has been considered in its original position by determining the contextual window and word length of every lexical word in the *corpus*. The word level contextual window comprises of immediately preceding/previous token of every current lexical word or token, immediate first successive token and immediate second successive token of every current lexical word or token of the dataset.

For tagging of different grammatical categories in the MM-POST dataset, 33 POS tags of the CLE tagset as provided in Table 2 have been used. The number of available POS tags in the CLE tagset are originally 35 but the MM-POST dataset has occurrences for 33 POS tags among them. The CLE Tagset contains 12 main linguistic categories and 35 subcategories. These 35 subcategories form the set of POS tags. However, in the MM-POST dataset, two of the categories including common particle (POS Tag: PRT) and common symbol (POS Tag: SYM) has no single occurrence in the news articles of the MM-POST dataset. The POS tag-wise frequency distribution of the MM-POST dataset is given in Table 5.

The actual instances of text for Urdu POS tags occurring in the MM-POST dataset are given as illustrative examples in Fig. 2.

### Features

The input features used for training and testing of the model are described as follows:

| Feature | Description |
| --- | --- |
| 1. **Word:** | the current word/token |
| 2. **PrevWord:** | previous word of the current word/token |
| 3. **NextWord:** | next word of the current word/token |
| 4. **Next2Word:** | second next word of the current word/token |
| 5. **WordLength:** | length of the current word/token |

The "Word" *i.e.*, the current word or token and the context words window of the current "Word" including "Previous Word", "Next Word" and "Second Next2 Word" are

**Table 4  MM-POST dataset.**

| Domain | Sentences | Tokens |
|---|---|---|
| Entertainment | 459 | 19,792 |
| Finance | 351 | 13,377 |
| General | 389 | 15,035 |
| Health | 430 | 16,084 |
| Politics | 579 | 27,409 |
| Science | 388 | 16,727 |
| Sports | 275 | 10,852 |
| Total | 2,871 | 119,276 |

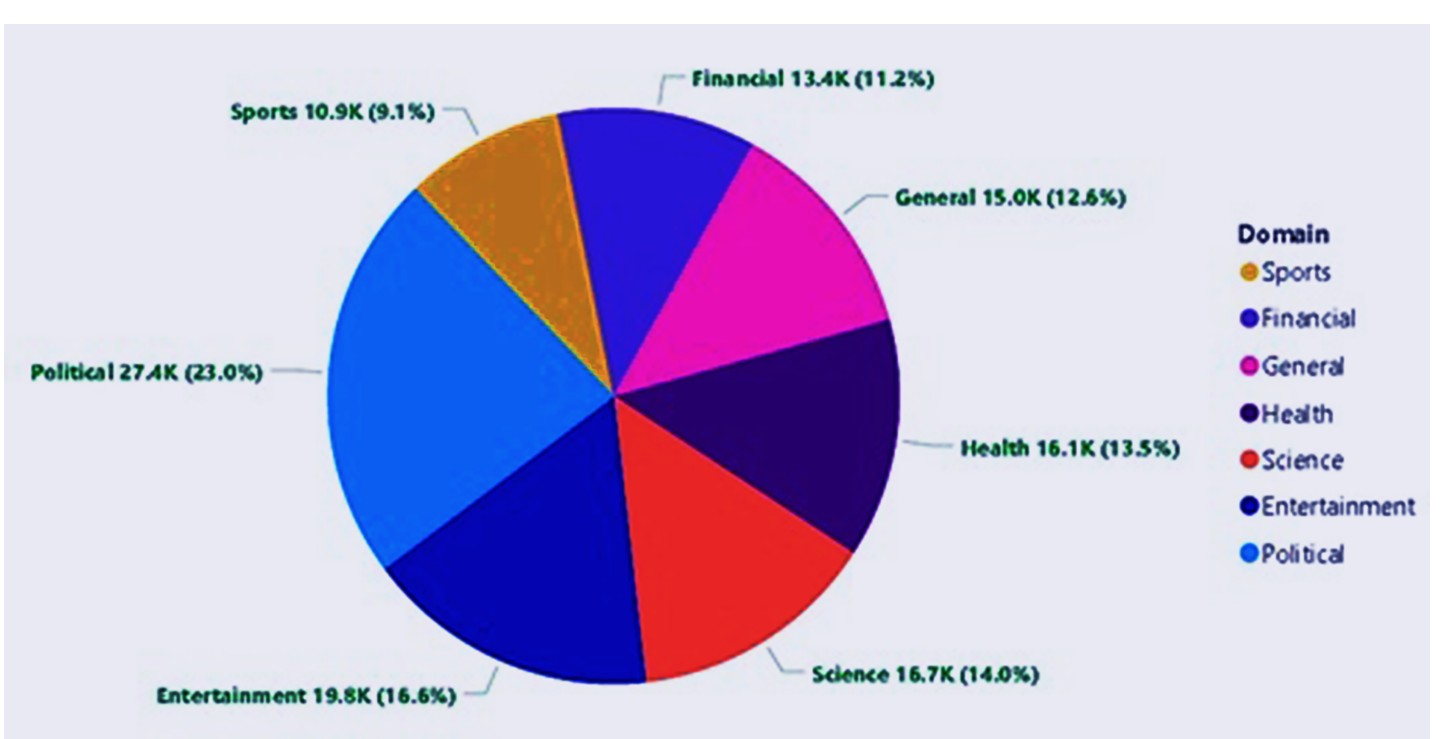

**Figure 1  Domain-wise distribution of POS tags in the MM-POST dataset.**

used together with the "Word Length" of the current "Word" as input features for learning the pattern and predicting the POS tag of every current word/token of the *corpus*.

The "Word" feature is the key feature that contains all the tokens including both words and other than words, for which the model has to learn the pattern and predict its corresponding POS tag. The "PrevWord" *i.e.*, Previous Word, is the word/token just preceding the current word/token. The "NextWord" *i.e.*, Next Word is the feature that contains the words/tokens just after the current word and the "Next2Word" *i.e.*, the second next word of the current word/token contains the 2nd next word/token of the current

**Table 5 Frequency of POS tags in the MM-POST dataset.**

| POS tag | Freq: | POS tag | Freq: | POS tag | Freq: |
|---|---|---|---|---|---|
| NN | 32,678 | AUXA | 2,325 | OD | 488 |
| PSP | 21,975 | VBI | 2,041 | VALA | 470 |
| VBF | 10,200 | CD | 1,930 | SCK | 367 |
| NNP | 8,622 | Q | 1,560 | PRS | 356 |
| PU | 7,052 | RB | 1,080 | PRD | 111 |
| JJ | 6,122 | NEG | 1,071 | PRF | 85 |
| PRP | 4,804 | PRR | 857 | INJ | 44 |
| AUXT | 4,370 | SCP | 635 | FR | 42 |
| CC | 3,042 | APNA | 629 | PRE | 38 |
| SC | 2,620 | AUXP | 551 | QM | 20 |
| PDM | 2,534 | AUXM | 544 | FF | 13 |
| Total | | | | 119,276 | |

word. Similarly, the "WordLength" (*i.e.*, Word Length), is the feature containing the length of words/tokens in terms of the number of characters of each current word/token.

The use of "Word", "Previous Word", "Next Word" and "Next2 Word" provides an effective utilization of these features as the context-window for every current word/token, in the prediction of POS tag for every current "Word" or token. One previous adjacent token (*i.e.*, the previous lexical word or token) and two next adjacent tokens (*i.e.*, next and second next lexical word or token) of the current word/token are considered to serve as the context window together with the Word Length of the current word/token in characterizing the target POS tag for every current word/token. The format of the training file is shown in Fig. 3.

The names of the input features *i.e.*, "Word", "PrevWord", "NextWord", and "Next2Word" contain the term "word" to keep them descriptive about the data they contain *i.e.*, the lexical words or tokens, although other tokens like numbers, punctuation, and special characters also exist. For example, the "Word" feature indicates the current word for which the POS tag is to be predicted but the individual values of this feature may also include other tokens like numbers and punctuation in addition to the Urdu textual words.

Different experimentation was made for inclusion and exclusion of various words both in proceeding and succeeding position in the context window. Increasing or decreasing the size of context window and selecting other words for the presently selected features did not improve results and even resulted in degraded performance.

## Experiment

The CRF model with Python-CRFsuite has been used for learning the pattern of the input features in predicting the POS tags in the MM-POST Urdu *corpus*. Python-CRFsuite provides an interface to the CRFsuite which is a CRF library implemented in C++. It allows the use of CRF model functionality within the Python scripts. CRF is well suited for sequence labeling tasks like POS tagging that involve sequential nature, context

| POS Tag | Illustrative Examples |
|---------|----------------------|
| NN | آرتھوپیڈک, استفسار, اکاؤنٹس, استفسارات, اکثریت, انتہا, اکثریتی, اٹھوں, آجائیں, استاد |
| NNP | پاکستان, انڈیا, جناح, خان, علی, محمد, نصرت, پولیو کیپاس, افغانستان |
| VBI | کرنے, کہنا, ہونے, کرنا, آنے, جانے, بنانے, رکھنے, ہونا, دینے |
| VBF | ہے, ہو, کیا, کر ہیں, کہا کرتے, کی, تھا, تھے, بتایا, تھی, ہوا, ہوتا, ہوتی, ہوئی, آ, ہوں, کہتے, لے, رہے |
| AUXA | گیا, ہوئے, جاتا, گئی, جائے, دیا, گئے, جا, جاتی, دی, جاتے, جانے, جائیں, لئیں, جائیں, ہوا چلی, ہوا چکے, لی, چکا |
| AUXP | رہے, رہی, رہا, رہیں, رہتے, رہتی, رہوں |
| AUXT | ہے, ہیں, تھا, تھے, گا, تھی, گے, گی, ہوں, تھیں, ہو |
| AUXM | سکتا, سکتی سکتے, سکے, چاہیے, سکیں, چاہتے, ہو, چاہتی, چاہتا, سکا, سکی, ہوں, رہتا, رہتی, سکتیں, چاہتیں |
| PRP | ان, اس, وہ, انھوں, اٹھیں, میں, ہم, آپ, اسے, مجھے, ہمیں, کسی, یہ, اُن, کون, کہاں, مجھ, انہوں, اسی, تم |
| PDM | یہ, اس, کوئی, کسی, ایسا, اسی, ایسی, ایسے, یہی, کس, ان, کن, وہی, اُس, لیسی, اُسی, انہی, اُن, فلاں, وہ |
| PRS | میرے, ہمارے, میری, ہماری, میرا, ہمارا, آور, تمہارے, تیری, تیرا تیرے |
| PRD | جو, جس, جن |
| PRR | جو, جس, جن, جیسے, جسے, جیسا, جٹھیں جیسی, جٹھوں, جنہیں, اتنی, جسے |
| PRF | خود |
| APNA | اپنے, اپنی, اپنا |
| JJ | سچا, دشوار, ترش, احمق, دفتری, سرسبز, سرفہرست, دلکش, بے جا, پکا |
| Q | زیادہ, بہت, کچھ, سب, کم, کئی, تمام, چند, اتنی, کافی, متعدد, انتہا, اکثر, اتنے, بعض, کتنا, کتنی, کتنے, جتنا |
| CD | ایک, دو, کروڑ, لاکھ, ہزار, ارب, سو, ملین, , صد, سوا |
| OD | دونوں, دوسرے, دوسری, پہلی, دوسرا, تیسرے, پہلا, فرسٹ, واحد, تینوں, چوتھے, تیسری, پہلے, سینکڑوں |
| FR | ساڑھے, نصف, ڈھائی, پونے, آدھی, آدھے, ڈیڑھ, نیم, آدھا |
| QM | گنا, دگنی, دگنا |
| RB | کیا, پھر, کیسے, کیوں, شاید, باوجود, متعلق, بطور, بالکل, دوبارہ, تقریباً, ہمیشہ, جلد, ضرور, سمیت |
| Neg | نہیں, نہ, نا, مت |
| PRE | فی, بغیر, دریں, سوائے |
| PSP | کے, میں, کی, سے, کا, نے, کوپر بھی, لیے, تک, بارے, علاوہ, آف, از, بغیر, بجائے, پہ, سوا, گئے, تا, خاطر, در |
| CC | اور, یا, لیکن, مگر, جبکہ, و, یعنی, بلکہ, اینڈ, حالانکہ, چنانچہ, گویا |
| SC | کہ, تو, کیونکہ, تاکہ, پھر, ورنہ, لہذا چونکہ, لہذا, بشرطیکہ, پر |
| SCK | کر, کے |
| SCP | اگر, لیکن, تاہم, مگر, اگرچہ, جبکہ, یعنی, پھر, بلکہ, البتہ, پس, چنانچہ, چونکہ |
| INJ | اے, خبردار, جی, کاش, او, ہاں, بابا, ارے, اچھا, |
| VALA | والے, والی, والا, والوں |
| PU | ۔, ؛, ' ' , ؟, ؛ , ( ), - |
| FF | اے, بی, این, ڈی, ان, جی, جی ایس کے, اے بی سی, آر, ایس |

**Figure 2  Illustrative examples of POS tags in the MM-POST dataset.**

| Word | PrevWord | NextWord | Next2Word | WordLength | POS |
|---|---|---|---|---|---|
| دل | ، | دل | پاکستان | 2 | NN |
| دل | دل | پاکستان | ، | 2 | NN |
| پاکستان | دل | ، | ایسا | 7 | NNP |
| ، | پاکستان | ایسا | ملی | 1 | PU |
| ایسا | ، | ملی | گیت | 4 | PDM |
| ملی | ایسا | گیت | ربا | 3 | JJ |
| گیت | ملی | ربا | جس | 3 | NN |
| ربا | گیت | جس | نے | 3 | VBF |
| جس | ربا | نے | مقبولیت | 2 | PRR |
| نے | جس | مقبولیت | کے | 2 | PSP |
| مقبولیت | نے | کے | ریکارڈ | 7 | NN |
| کے | مقبولیت | ریکارڈ | توڑے | 2 | PSP |
| ریکارڈ | کے | توڑے | ۔ | 6 | NN |
| توڑے | ریکارڈ | ۔ | آج | 4 | NN |
| ۔ | توڑے | آج | سے | 1 | PU |

**Figure 3  CRF POS tagging training-file example.**

dependency, and labeling of each word. The input variables used are "Word", "PrevWord", "NextWord", "Next2Word" and "WordLength" for learning and prediction of the labels of the target variable "POS" *i.e.*, the Part of Speech tag.

The MM-POST dataset was split into 80% Training portion and 20% Testing. The part of the dataset considered for the training portion incorporated 95,420 tokens and the testing portion contained 23,855 tokens. The data of the dataset is already in a tokenized format of words/tokens, therefore there is no need for tokenization of text. The dataset was imported from a Microsoft Excel file. Each column of the file represents a feature and each row of the file describes a record to be input to the model. The five columns on the left side are input features whereas the rightmost sixth column is the target variable of the training file. However, in the testing phase, the target variable is not included in the input of the model and the model performs prediction of labels based on the learning achieved during its training.

The CRF model has been used with "lbfgs" as an optimization algorithm. The maximum number of iterations for the algorithm to reach the optimized result is kept at 100 with the

inclusion of all possible label transitions. The optimization algorithm with its default 100 iterations for the CRF model was chosen because of empirical testing with convergence speed and memory efficiency. The model showed a balance between accuracy and Overfitting for the "lbfgs" value of 100. For higher values, the unnecessary computational overhead was observed. The model is trained over the training data to learn the patterns and relationships from input features within the data for predictions of 33 POS tags as labels that exist in the "POS" target variable.

# EVALUATION OF RESULTS

## Performance metrics

The performance of the CRF model has been measured using the evaluation metrics including precision, recall, F1-measure, and accuracy. The Table 6 shows the values for these evaluation metrics in addition to support values for all the target labels *i.e.*, the Urdu POS tags. Hence the dataset is split into 80% training and 20% testing portions, the Support value for every POS tag is the number of samples of every POS tag or class-label existing in the testing portion. For every class-label of the dataset, 80% of the samples have been selected to become part of the training set and 20% of the testing set. The Support value for every label is the count of 20% of the total number of instances of a particular class-label that has been included in the testing data. For example, the total frequency of proper noun (NNP) is 8,622 in the dataset. The 6,900 NNP tokens have been included in the training data whereas 1,720 tokens have been made part of the testing data. Thus, the support value for the class-label NNP is 1,722. The overall Support for the dataset having 119,276 tokens is 23,855.

A brief description and formulas of the evaluation metrics are given as below:

## Precision

Precision measures how correctly the model tags the words. It is helpful particularly in understanding the assignment of POS tags to frequent words or when incorrect tagging has been resulted for larger instances of words. High precision means reducing false positive predictions of the model.

$$\text{Precision} = \frac{\text{True Positives}}{\text{True Positives} + \text{False Positives}} \qquad (2)$$

## Recall

Recall ensures that the model captures most of the instances of words of a particular POS class/tag. It is the ratio of all correctly predicted/tagged words to all actual tags of words. High recall attempts lowering the number of false negatives, and it reflects the model ability to correctly predict most instances of a class.

$$\text{Recall} = \frac{\text{True Positives}}{\text{True Positives} + \text{False Negatives}} \qquad (3)$$

**Table 6 Evaluation metrics—CRF Urdu POS tagging.**

| POS | Precision | Recall | F1-score | Support |
|---|---|---|---|---|
| APNA | 1 | 0.99 | 1 | 120 |
| AUXA | 0.95 | 0.95 | 0.95 | 459 |
| AUXM | 0.99 | 0.94 | 0.96 | 113 |
| AUXP | 0.93 | 0.94 | 0.93 | 95 |
| AUXT | 0.98 | 0.97 | 0.98 | 868 |
| CC | 0.98 | 0.99 | 0.99 | 643 |
| CD | 0.97 | 0.97 | 0.97 | 359 |
| FF | 1 | 0 | 0 | 7 |
| FR | 0.88 | 0.7 | 0.78 | 10 |
| INJ | 1 | 0 | 0 | 8 |
| JJ | 0.97 | 0.89 | 0.93 | 1,241 |
| NEG | 1 | 0.99 | 0.99 | 226 |
| NN | 0.92 | 0.98 | 0.95 | 6,550 |
| NNP | 0.94 | 0.86 | 0.9 | 1,722 |
| OD | 1 | 0.89 | 0.94 | 105 |
| PDM | 0.98 | 0.98 | 0.98 | 485 |
| PRD | 1 | 0.76 | 0.86 | 21 |
| PRE | 1 | 0.71 | 0.83 | 7 |
| PRF | 1 | 1 | 1 | 23 |
| PRP | 0.99 | 0.97 | 0.98 | 984 |
| PRR | 0.97 | 0.98 | 0.97 | 174 |
| PRS | 1 | 0.98 | 0.99 | 81 |
| PSP | 0.99 | 1 | 0.99 | 4,320 |
| PU | 1 | 1 | 1 | 1,444 |
| Q | 1 | 0.98 | 0.99 | 298 |
| QM | 1 | 0.33 | 0.5 | 3 |
| RB | 0.96 | 0.91 | 0.93 | 211 |
| SC | 1 | 0.99 | 0.99 | 504 |
| SCK | 0.9 | 0.93 | 0.92 | 71 |
| SCP | 0.97 | 0.85 | 0.91 | 123 |
| VALA | 0.99 | 1 | 1 | 109 |
| VBF | 0.94 | 0.93 | 0.94 | 2,073 |
| VBI | 0.98 | 0.92 | 0.95 | 398 |
| Accuracy | | | 0.96 | 23,855 |
| Macro avg | 0.98 | 0.86 | 0.88 | 23,855 |
| Weighted avg | 0.96 | 0.96 | 0.96 | 23,855 |

## F1-score

F1-score combines precision and recall, offering a unified metric for performance. F1 is the key metric when both the false positives and false negatives are important, or POS tags are

unevenly distributed. High F1-score indicates that the model is accurately predicting tags for the words and identifying all instances of a POS class/tag.

$$\text{F1-score} = \frac{2 \times \text{Precision} \times \text{Recall}}{\text{Precision} + \text{Recall}} \tag{4}$$

## Accuracy

Accuracy reflects the model's overall ability to correctly tag or assign labels to words across all POS categories. It is a good metric for knowing the percentage correct prediction of a model but only in balanced datasets because for unbalanced data, the accuracy can be less informative.

$$\text{Accuracy} = \frac{\text{Number of Correct Predictions}}{\text{Total Number of Predictions}} \tag{5}$$

## Results analysis

The bottom rows of Table 6 show the values for accuracy, macro average, and weighted average. Accuracy means the overall aggregated number of correct predictions of POS tags per total number of predictions of POS tags. The overall accuracy achieved for the CRF model is reported as 96.1%.

The macro average metrics are used to evaluate the model performance across all classes treating them equally and are calculated by taking the simple average of results for all the classes, giving equal weight to the result of each class regardless of its size in the actual data. The macro average values for precision, recall, and F1-score of our CRF model are 0.98, 0.86, and 0.88 respectively. The weighted average metrics are used to know the model performance based on influencing the result on class distribution *i.e.*, giving more importance to larger classes. The disparity between the high macro-precision (0.98) and lower recall (0.86) indicates that the model has been successful in reducing the False Positives by most of the times (98%) correctly predicting the tag for a word and in very few cases it misclassified them to incorrect tag. The recall of 0.86 means that the model misses to avoid false negatives in some cases *i.e.*, for certain POS class, the model fails to recognize the correct tag for the words. The lower recall can be improved by enabling the model training over sufficiently large instances of the rare class labels or POS tags. Thus, the pattern for missed classified instances of the present dataset shall be properly learned by the model and performance shall be further improved.

The weighted result is achieved for every class by its support value and the sum of the weighted values. The weighted average value of our CRF model for accuracy, precision, recall, and F1-measure each, is 0.96. It means that after giving effect to the larger classes to influence the model in predictions, the performance metrics improve substantially which can be seen through the enhancement of the F1-score from 0.88 (macro average) to 0.96 (weighted average). Higher values reported for weighted average than macro average for the performance metrics of our model highlight the need to have a sufficiently larger

number of occurrences for all the label classes and enhancing the number of observations or samples for the low-frequency classes shall improve the effectiveness of the model.

Out of the total 33 POS tags, 26 (*i.e.*, 79% of labels) have an F1-score of more than 0.90 which is encouraging regarding the efficiency of the model. The four POS tags have an F1-score of more than 0.78, one POS tag (*i.e.*, QM) has an F1-score of 0.50, and two (*i.e.*, FF and INJ) have zero F1-score. The zero F1-score for the two POS tags FF and INJ is due to the lower Support values of only seven and eight respectively. The two POS Tags "FF" and "INJ" have rare frequency of only seven and eight respectively in the dataset. The "FF" tag has been confused six times with "NN" and once with "PRP" as is shown in the Confusion Matrix of Fig. 4. The "INJ" tag has been confused four times with "NN", two times with "NNP", once with "CD" and once with "PU". Thus, the rare occurrence frequencies of both the POS Tags "FF" and "INJ" caused lack of required contextual understanding for the model in their tagging. Similarly, the QM POS tag has a support value of only three.

Most of the POS tags have been correctly predicted to their corresponding true tags as demonstrated in the Confusion Matrix of Fig. 4. The Confusion Matrix reflects that the "NN" tag has been confused the most with "NNP" & "VBF". The "NNP" has been wrongly predicted the most as "NN" and in a few cases as "JJ". The "VBF" has been confused the most with "NN" & "AUXA", the "PSP" is confused in a few cases with "NN" & "VBF" and the label "JJ" is confused the most with "NN", "CD" & "VBF".

The number of True predictions both for positive and negative cases are higher for most of the labels as is evident from the figures for true positives (TP), false positives (FP), true negatives (TN), and false negatives (FN) for the POS tags of the dataset in Table 7.

## Generalization on external data

To evaluate the generalization capability of our trained model on external validation data, an unseen news article of 1,161 words or tokens that is not part of the MM-POST dataset, was used. The data was converted into the trained model's format and the saved model was loaded. The POS tagging label-prediction was performed through the model. Analysis of the results revealed that out of 1,161 words or tokens, the model correctly predicted 1,134 words or tokens (97.7%) whereas 27 tokens (2.3%) were labelled incorrectly. The overall accuracy achieved for the external validation data of 97.7% is highly encouraging. The average accuracy resulted for an individual POS tag becomes 83% that is less than the average accuracy achieved by the model for training dataset (*i.e.*, 88%). This is due to far less number of instances of individual labels in the external validation data than the training dataset, in addition to the model failure in correct prediction of unseen distinctive instances.

Interestingly, the incorrect predictions resulted for only 10 out of 33 POS tags or labels whereas correct predictions for the remaining 22 POS tags were resulted. Six among the POS tags had incorrect predictions for 23 times and the remaining four have one incorrect prediction each. The label-wise accuracy achieved by the trained model for external validation data is given in Table 8.

Thus the outstanding performance of the trained CRF model in POS tagging of training dataset as well unseen external validation data, demonstrates the model ability of generalization to the external out-of-sample data and proves scale-able to unseen data.

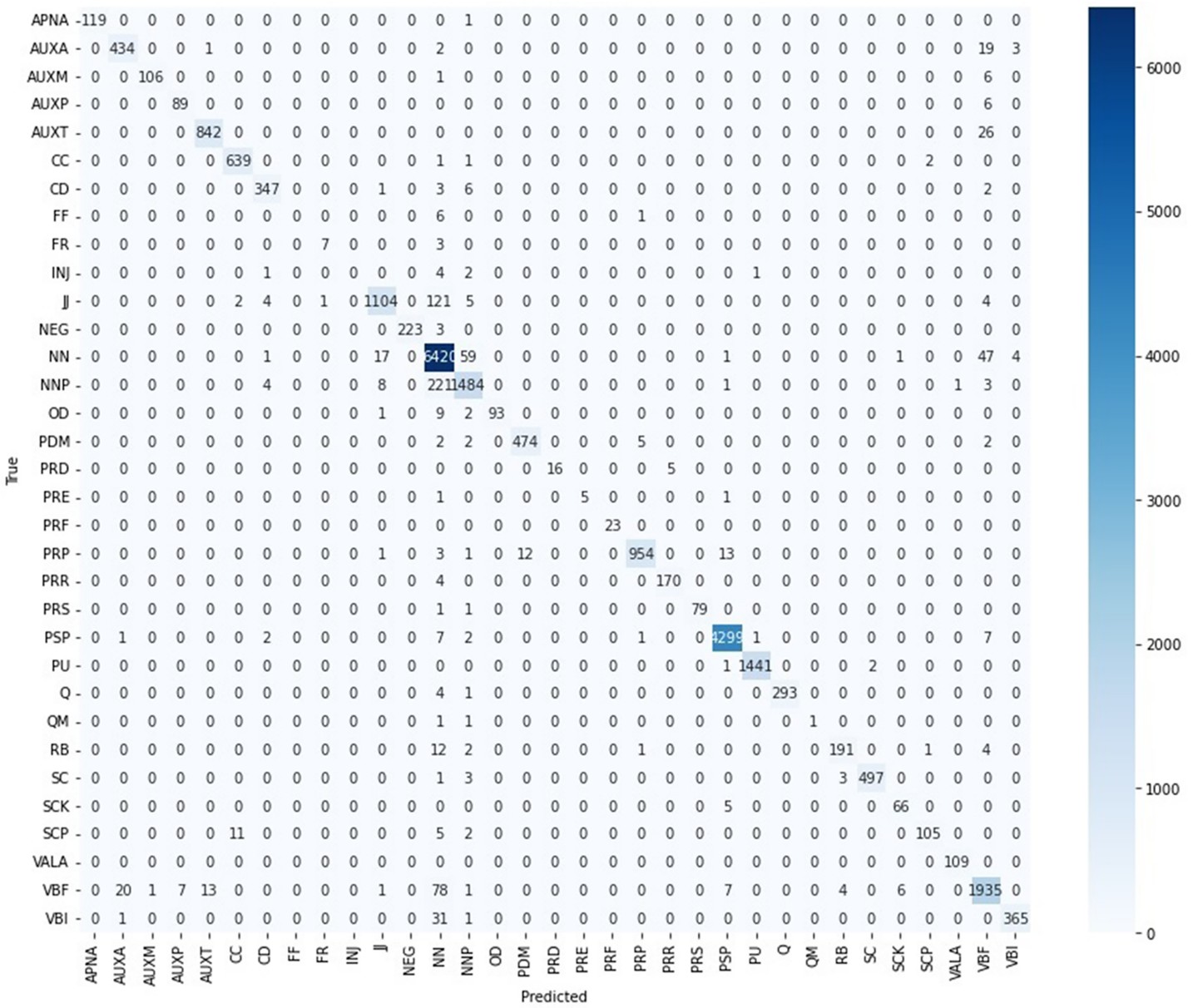

**Figure 4 Confusion matrix—CRF Urdu POS tagging.**               

## Implementation of proposed CRF model using Urdu universal dependency treebank

The implementation and evaluation of our proposed model were performed using POS tagged data from Urdu Universal Dependency Treebank (UDTB). The UDTB was developed at IIIT Hyderabad India by automatic conversion from Urdu Dependency Treebank (*Bhat et al., 2017*). The data containing 14,604 Urdu words tagged with 17 POS tags downloaded from *Bhat & Zeman (2024)* was used for training and testing of our proposed CRF based supervised POS classifier. For easy comparison, compound Urdu

**Table 7 Confusion matrix—CRF POS tagging.**

| POS | TP | FP | TN | FN | Support |
|---|---|---|---|---|---|
| APNA | 119 | – | 23,735 | 1 | 120 |
| AUXA | 434 | 22 | 23,374 | 25 | 459 |
| AUXM | 106 | 1 | 23,741 | 7 | 113 |
| AUXP | 89 | 7 | 23,753 | 6 | 95 |
| AUXT | 842 | 14 | 22,973 | 26 | 868 |
| CC | 639 | 13 | 23,199 | 4 | 643 |
| CD | 347 | 12 | 23,484 | 12 | 359 |
| FF | – | – | 23,848 | 7 | 7 |
| FR | 7 | 1 | 23,844 | 3 | 10 |
| INJ | – | – | 23,847 | 8 | 8 |
| JJ | 1,104 | 29 | 22,585 | 137 | 1,241 |
| NEG | 223 | – | 23,629 | 3 | 226 |
| NN | 6,420 | 524 | 16,781 | 130 | 6,550 |
| NNP | 1,484 | 93 | 22,040 | 238 | 1,722 |
| OD | 93 | – | 23,750 | 12 | 105 |
| PDM | 474 | 12 | 23,358 | 11 | 485 |
| PRD | 16 | – | 23,834 | 5 | 21 |
| PRE | 5 | – | 23,848 | 2 | 7 |
| PRF | 23 | – | 23,832 | – | 23 |
| PRP | 954 | 8 | 22,863 | 30 | 984 |
| PRR | 170 | 5 | 23,676 | 4 | 174 |
| PRS | 79 | – | 23,774 | 2 | 81 |
| PSP | 4,299 | 29 | 19,506 | 21 | 4,320 |
| PU | 1,441 | 2 | 22,409 | 3 | 1,444 |
| Q | 293 | – | 23,557 | 5 | 298 |
| QM | 1 | – | 23,852 | 2 | 3 |
| RB | 191 | 7 | 23,637 | 20 | 211 |
| SC | 497 | 2 | 23,349 | 7 | 504 |
| SCK | 66 | 7 | 23,777 | 5 | 71 |
| SCP | 105 | 3 | 23,729 | 18 | 123 |
| VALA | 109 | 1 | 23,745 | – | 109 |
| VBF | 1,935 | 126 | 21,656 | 138 | 2,073 |
| VBI | 365 | 7 | 23,450 | 33 | 398 |

words were broken into single words and few POS tags of the UDTB data were renamed to the CLE Tagset used for tagging of the MM-POST dataset.

The model achieved an accuracy of 89.6% using UDTB dataset in comparison to the accuracy of 96.1% for MM-POST dataset. The results show that using the UDTB dataset having eight times less number of POS tagged tokens than the MM-POST dataset, the performance of the model degraded only to approximately six percent. This demonstrates the scalability and generalizability of our CRF-based model for Urdu POS tagging on a

**Table 8 Evaluation of trained CRF model on external validation data.**

| POS tag | Correct prediction | Incorrect prediction | Total | Accuracy |
|---------|--------------------|--------------------|-------|----------|
| NN | 325 | 5 | 330 | 98.5 |
| PSP | 202 | | 202 | 100 |
| VBF | 104 | 2 | 106 | 98.1 |
| PU | 62 | | 62 | 100 |
| PRP | 61 | 4 | 65 | 93.8 |
| JJ | 57 | 7 | 64 | 89.1 |
| AUXT | 49 | 1 | 50 | 98.0 |
| PDM | 36 | 1 | 37 | 97.3 |
| NNP | 33 | 2 | 35 | 94.3 |
| CC | 30 | | 30 | 100 |
| SC | 26 | 1 | 27 | 96.3 |
| VBI | 26 | 3 | 29 | 89.7 |
| AUXA | 18 | | 18 | 100 |
| NEG | 16 | | 16 | 100 |
| PRR | 14 | | 14 | 100 |
| Q | 11 | | 11 | 100 |
| AUXP | 10 | | 10 | 100 |
| CD | 10 | | 10 | 100 |
| RB | 9 | | 9 | 100 |
| SCK | 9 | | 9 | 100 |
| PRF | 5 | | 5 | 100 |
| OD | 5 | 1 | 6 | 83.3 |
| VALA | 5 | | 5 | 100 |
| PRD | 4 | | 4 | 100 |
| PRS | 3 | | 3 | 100 |
| APNA | 2 | | 2 | 100 |
| SCP | 1 | | 1 | 100 |
| AUXM | 1 | | 1 | 100 |
| Total | 1,134 | 27 | 1,161 | 97.7 |

smaller dataset, generated from different genres and annotated with a tagset different from the CLE tagset that we primarily used. The evaluation metrics given in Table 9 show the POS tag-wise results of the CRF model achieved using POS tagged Urdu UDTB data.

## SVM implementation and comparison with CRF-based Urdu POS tagging

The SVM model has been widely used in the research community for Urdu POS tagging. The implementation of our proposed approach using SVM model was made with our selected features set (Word, PrevWord, NextWord, Next2Word, WL) and the MM-POST dataset. The SVM model achieved an accuracy of only 68% which is far less than the accuracy of 96% achieved by the CRF. The results given in Table 10 show that only three

**Table 9 Evaluation metrics—CRF Urdu POS tagging using UDTB.**

| POS | Precision | Recall | F1-score | Support |
|---|---|---|---|---|
| AUXT | 0.33 | 0.50 | 0.40 | 2 |
| CC | 0.98 | 0.99 | 0.98 | 136 |
| JJ | 0.81 | 0.79 | 0.80 | 261 |
| NEG | 1.00 | 1.00 | 1.00 | 10 |
| NN | 0.84 | 0.93 | 0.88 | 785 |
| NNP | 0.82 | 0.73 | 0.78 | 278 |
| PDM | 0.95 | 0.80 | 0.87 | 50 |
| PRP | 0.92 | 0.86 | 0.89 | 102 |
| PSP | 0.98 | 0.98 | 0.98 | 586 |
| Q | 0.94 | 0.82 | 0.88 | 102 |
| RB | 1.00 | 0.23 | 0.38 | 13 |
| RP | 1.00 | 0.84 | 0.91 | 37 |
| SYM | 1.00 | 1.00 | 1.00 | 111 |
| VAUX | 0.91 | 0.90 | 0.91 | 186 |
| VBF | 1.00 | 0.50 | 0.67 | 4 |
| VM | 0.91 | 0.88 | 0.90 | 249 |
| Punct | 1.00 | 1.00 | 1.00 | 12 |
| Accuracy | | | 0.90 | 2,924 |
| Macro avg | 0.91 | 0.81 | 0.84 | 2,924 |
| Weighted avg | 0.9 | 0.9 | 0.89 | 2,924 |

out of 33 POS tags (*i.e.*, AUXT, PSP and PU) have F1-score as 0.80 or above. For all the others the SVM has failed in correct tagging. The analysis reveals that SVM is unable to properly learn and classify most of the POS tags except few because it could not successfully model the contextual window information of the lexical word and their corresponding POS tags. Thus, proving our CRF based model performing better and suitable in the sequence labelling task of Urdu POS tagging.

## Comparison with benchmark approaches

Our approach for CRF model implementation achieved an accuracy of 96.1% which is higher than the CRF accuracy of 88.74% by *Khan et al. (2019b)*, the 83.52% on CLE dataset by *Khan et al. (2019a)*, the 88.4% on Bushra Jawaid dataset by *Khan et al. (2019a)* and the accuracy of 95.8% on Bushra Jawaid dataset by *Nasim, Abidi & Haider (2020)*. However, the performance achieved by *Nasim, Abidi & Haider (2020)* using the CRF together with BiLSTM (*i.e.*, 96.3%) is subtly higher than our approach by 0.2%.

The benchmark approaches used large sized feature sets making them complex to understand and computationally less efficient as detailed in the following:

*Khan et al. (2019b)* used language-dependent (*i.e.*, POS tag of the previous word and suffix of the current word) and language-independent features (*i.e.*, context words window). They used ten unigram templates for feature set generation. Their features set

**Table 10 Evaluation metrics—SVM implementation of Urdu POS tagging.**

| POS tag | Precision | Recall | F1-score | Support |
|---|---|---|---|---|
| APNA | 0.47 | 0.49 | 0.48 | 122 |
| AUXA | 0.66 | 0.71 | 0.69 | 444 |
| AUXM | 0.69 | 0.73 | 0.71 | 105 |
| AUXP | 0.75 | 0.82 | 0.78 | 104 |
| AUXT | 0.76 | 0.85 | 0.80 | 860 |
| CC | 0.59 | 0.66 | 0.62 | 655 |
| CD | 0.49 | 0.57 | 0.53 | 389 |
| FF | 1.00 | 0.50 | 0.67 | 2 |
| FR | 0.45 | 0.56 | 0.50 | 9 |
| INJ | 0.33 | 0.27 | 0.30 | 11 |
| JJ | 0.42 | 0.34 | 0.38 | 1,229 |
| NEG | 0.50 | 0.59 | 0.54 | 215 |
| NN | 0.67 | 0.72 | 0.69 | 6,621 |
| NNP | 0.72 | 0.66 | 0.69 | 1,722 |
| OD | 0.39 | 0.46 | 0.42 | 93 |
| PDM | 0.45 | 0.40 | 0.42 | 498 |
| PRD | 0.24 | 0.31 | 0.27 | 16 |
| PRE | 0.00 | 0.00 | 0.00 | 4 |
| PRF | 0.56 | 0.50 | 0.53 | 18 |
| PRP | 0.55 | 0.46 | 0.50 | 972 |
| PRR | 0.53 | 0.57 | 0.55 | 147 |
| PRS | 0.27 | 0.22 | 0.24 | 78 |
| PSP | 0.78 | 0.84 | 0.81 | 4,361 |
| PU | 0.99 | 0.99 | 0.99 | 1,450 |
| Q | 0.43 | 0.34 | 0.38 | 307 |
| QM | 0.00 | 0.00 | 0.00 | 5 |
| RB | 0.40 | 0.29 | 0.34 | 218 |
| SC | 0.71 | 0.75 | 0.73 | 515 |
| SCK | 0.58 | 0.53 | 0.56 | 75 |
| SCP | 0.46 | 0.32 | 0.38 | 128 |
| VALA | 0.43 | 0.45 | 0.44 | 88 |
| VBF | 0.66 | 0.53 | 0.59 | 2,019 |
| VBI | 0.62 | 0.53 | 0.57 | 375 |
| Accuracy | | | 0.68 | 23,855 |
| Macro avg | 0.53 | 0.51 | 0.52 | 23,855 |
| Weighted avg | 0.67 | 0.68 | 0.68 | 23,855 |

included "Previous Lexical Word", "Current Lexical Word", "Next Lexical Word", "Current Lexical Word + Previous Lexical Word", "Current Lexical Word + Next Lexical Word", "Current Lexical Word + N-1 and N-2 Previous Words", "Current Lexical Word +
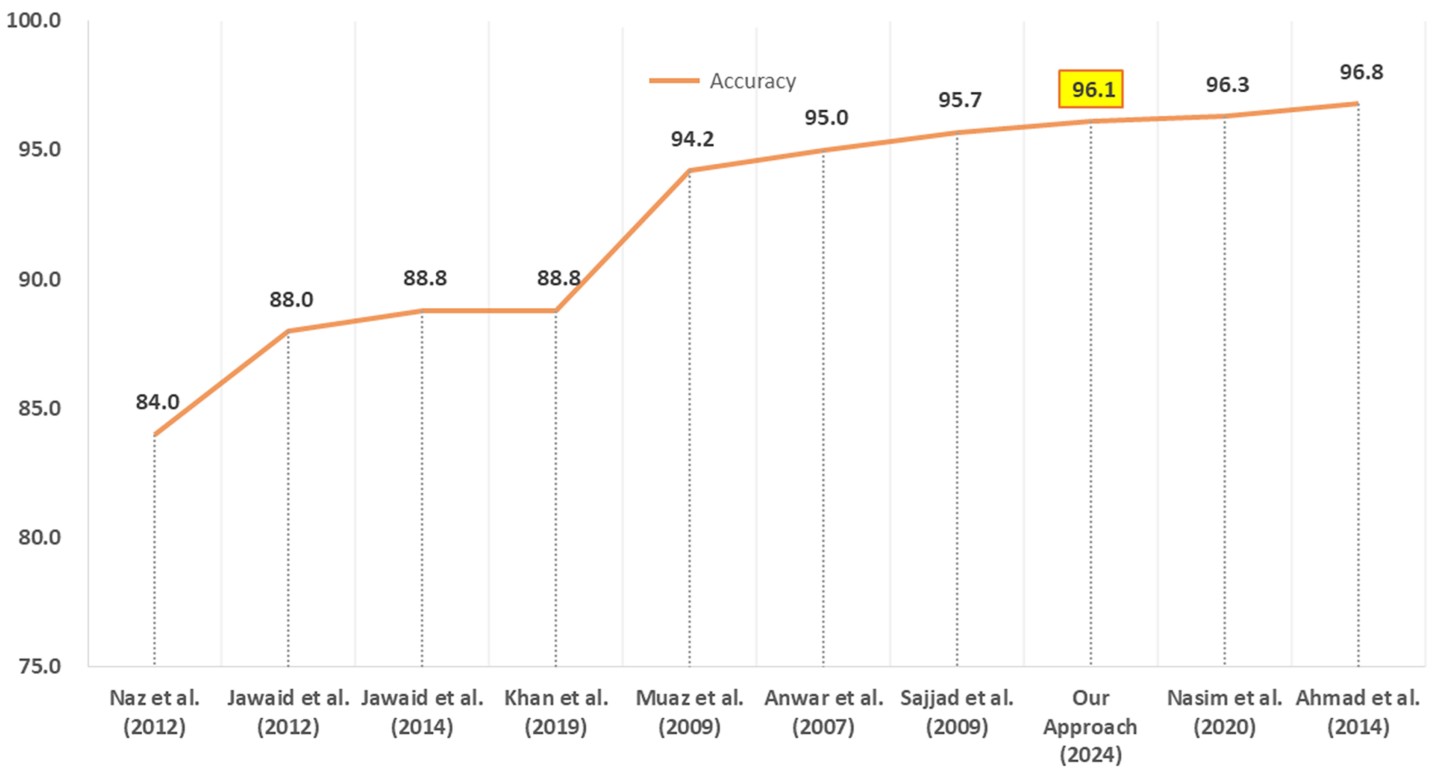

**Figure 5** Comparison of our CRF POS tagging approach with other approaches.

N+1 and N+2 Next Words", "Part of Speech tag of Previous Lexical Word", "Suffix of Current Lexical Word" and "Length of Current Lexical Word".

*Khan et al. (2019a)* used the context word features including (1) the token (the current word) (2) the word to the left of the current word (3) the word to the right of the current word (4) Joint use of the current word and the word to the left of the current word (5) Joint use of the current word and the word to the right of the current word (6) Joint use of the current word and N-1, N-2 left words of the current word, and (7) joint use of the current word and N+1, N+2 right words of the current word.

*Nasim, Abidi & Haider (2020)* utilized the features including Word, Length, Is_First, Is_Last, Suffix, Prev_Word_1, Prev_Word_2 and Next_Word of the current word.

Thus the use of simple and fewer language-independent features of Urdu text combined with the efficient performance make our CRF-based Urdu POS tagging approach surpassing the previous benchmark CRF approaches. Figure 5 provides a comparison of performance between our work and other researchers' approaches for Urdu POS tagging. The performance of our Urdu POS tagging approach is better than seven among nine researchers whereas two of them have slightly better accuracy.

In contrast to previous works, our Urdu POS tagging approach benefits from the small-sized feature set comprising only of five features; four among them are lexical word-based and one is the word length. These few word-based features in addition to the word length

are simple to determine and are language-independent. Thus enabling our Urdu POS tagging approach to have the potential for scaling, generalization, and adaptability.

## CONCLUSION AND FUTURE WORK

A CRF based automatic POS classifier for Urdu news text using the MM-POST dataset was discussed. The model achieved state-of-the-art performance by attaining an overall accuracy of 96.1% and macro average values for Precision, Recall, and F1-score as 98%, 86%, and 88% respectively using the training dataset. The trained model proved excellent ability of generalization by achieving even higher performance than on training dataset. The overall accuracy of 97.7% was reported for prediction of POS tags on external validation data that is not part of the training dataset. However, the average prediction accuracy of an individual POS tag on external validation data remained 83% in comparison to 86% on training data. It can be improved further by increasing the size of annotated data in the dataset for training of the model to learn further distinctive instances and variations of Urdu POS.

The input features "Word", "Previous Word", "Next Word" and "Second Next Word" of current word/token, used as context words window served well in addition to the "Word Length" feature of the current word/token in the classification and prediction of the Urdu POS tags. The utilization of lexical words as context window of current words helped in the effective learning and prediction of Urdu POS tags.

The CRF model has been proved efficient in multi-label or multi-class classification and predictions of Urdu POS tags for the dataset having 33 number of POS tags. The model achieved excellent performance for most of the POS tags, particularly for those having a sufficient number of occurrences in the dataset.

However, the performance can be further improved and the Weighted and Macro Average values of the F1-score can be enhanced from 0.96 and 0.88 respectively, if the size of the POS-tagged *corpus* is increased to incorporate a sufficiently large number of instances particularly for the less frequent POS tags; for example, "FF", "INJ" and "QM" having the support values of only seven, eight and three respectively. Thus the CRF model will be able to effectively predict the POS tags with high accuracy through the use of selected features set.

Our approach for Urdu POS tagging has the potential for expansion to other Indo-Aryan languages particularly which are agglutinative and free word order languages like Hindi, Punjabi, Pashto and Arabic. Experimentation can be done with the easy to determine and computationally efficient features set in other languages using CRF, other machine learning or deep learning models and modern ensemble or transformer-based models. The ease and effectiveness in selection and processing of the features set provides the opportunity of customization and introduction of further sophistication for all types of natural languages. Our work opens up avenues of research for application of proposed approach for other Urdu NLP tasks including named entity recognition, sentiment analysis, machine translation and text to speech systems, and so on.

In our future work, the POS tags generated through our presented approach, shall be employed as one of the features set for various prediction-tasks like named entity recognition of Urdu text using ML and deep learning techniques.

## ACKNOWLEDGEMENTS

We are grateful to our institute and our families & friends for their unwavering support and encouragement during the course of this research.

### Funding

The authors received no funding for this work.

### Competing Interests

The authors declare that they have no competing interests.

### Author Contributions

- Mushtaq Ali conceived and designed the experiments, performed the experiments, analyzed the data, performed the computation work, prepared figures and/or tables, authored or reviewed drafts of the article, and approved the final draft.
- Muzammil Khan conceived and designed the experiments, performed the experiments, analyzed the data, prepared figures and/or tables, authored or reviewed drafts of the article, and approved the final draft.
- Yasser Alharbi conceived and designed the experiments, authored or reviewed drafts of the article, and approved the final draft.

### Data Availability

The raw data and code are available in the Supplemental Files.

The Mushtaq and Muzammil Part of Speech Tagged (MM-POST) dataset is available at GitHub and Zenodo:

- https://github.com/Mushtaq-Ali/MM-POST-dataset.

- Mushtaq Ali. (2024). Mushtaq-Ali/MM-POST-dataset: MM-POST dataset v1.0.0 (POSTagging). Zenodo. https://doi.org/10.5281/zenodo.14165184.

The POS tagged data from Urdu Universal Dependency Treebank (UDTB) is available at GitHub: https://github.com/UniversalDependencies/UD_Urdu-UDTB/blob/master/README.md.

### Supplemental Information

Supplemental information for this article can be found online at http://dx.doi.org/10.7717/peerj-cs.2577#supplemental-information.

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
