# Peer review of "A conditional random field based approach for high-accuracy part-of-speech tagging using language-independent features"

_PeerJ Computer Science, doi:10.7717/peerj-cs.2577_

## Round 0.1 · original submission · Minor Revisions

Dear authors,

Thank you for submitting your manuscript. Feedback from the reviewers is now available. It is not recommended that your article be published in its current format. However, we strongly recommend that you address the issues raised by the reviewers and resubmit your paper after making the necessary changes. Note the PDF from Reviewer 3.

Before submitting the paper following should also be addressed:

1. More details on data preprocessing should be provided for improving the clarity and reproducibility
2. Equations should be used with correct equation number. Many of the equations are part of the related sentences. Attention is needed for correct sentence formation.
3. How "Support" values are computed should be provided like other evaluation metrics.
4. "...given in table 8...", "...In table 3, a...", "...as shown in table 4...", "..as provided in table 1...", "...is shown in table 5...", "...The table 6 show...", "...rows of table 6 show...", "...dataset in table 7..." should be corrected.
5. Please pay attention to the usage of abbreviation. Introduce every acronym before using it in the text. The first time you use the term, put the acronym in parentheses after the full term.

Best wishes,

·

Basic reporting

In this paper a Conditional Random Field (CRF) based supervised POS
classiûer has been developed for 33 diûerent Urdu POS categories using the language independent features of Urdu text for the Urdu news dataset MM-POST containing 119,276 tokens of seven diûerent domains, including Entertainment, Finance, General, Health, Politics, Science and Sports.

paper is well written and it can be published.

Experimental design

Urdu POS tagging through CRF has been used.
there is alraedy" https://github.com/UniversalDependencies/UD_Urdu-UDTB/blob/master/README.md";
tagger system are there.
u should use other evaluation metrics to justify your proof.

Validity of the findings

no

Additional comments

no

Reviewer 2 ·

Basic reporting

The paper is written in clear, professional, and unambiguous English throughout. There are no significant grammatical errors or awkward sentence structures, making the text accessible to readers. The language used is appropriate for an academic article.
The introduction provides a solid background on the problem of Part-of-Speech (POS) tagging, particularly for resource-scarce languages such as Urdu. The motivation for using CRF for POS tagging is well justified, emphasizing the challenges associated with Urdu text processing and how the proposed approach addresses these issues. The authors also reference relevant literature, but some additional references to contemporary work in the field of POS tagging, especially for low-resource languages, would strengthen the context.
The structure of the paper is clear and follows discipline norms.
The figures and tables provided are relevant and add value to the text.

Experimental design

The article's scope falls well within the aims of a journal focused on NLP and computational linguistics, particularly given its focus on POS tagging for a low-resource language like Urdu.
The methodology is clearly described, and sufficient detail is provided for replicating the experiments. The authors outline the dataset used (MM-POST), the selection of language-independent features, and the structure of the CRF model, making it straightforward for others to replicate the work. More details about the preprocessing steps, such as tokenization, would enhance clarity.
In discussing the feature selection (word-level features, word length, etc.), authors should expand on any preprocessing steps applied to the dataset, such as normalization.
The use of standard evaluation metrics like Precision, Recall, F1-Score, and Accuracy is appropriate.

Validity of the findings

The findings are valid, well-supported by the experimental results, the accuracy achieved (96.1%) is impressive.
The experiments performed are thorough, with results demonstrating the effectiveness of the CRF-based model in comparison to existing methods. The authors provide adequate justification for their model’s superiority.
The paper does identify some limitations, such as the zero F1-score for low-frequency POS tags, and acknowledges the need for a larger dataset to improve results further. However, the authors could expand on potential future directions, especially in terms of adapting their model to other low-resource languages (agglutinative) or incorporating more complex feature sets.

Additional comments

Additional references to recent works in low-resource language POS tagging would provide a stronger context for the novelty of the approach.
It would be useful to conduct experiments with augmented data to increase the data size, which would allow us to test how the CRF model acts to larger data sets. This could also provide a deeper understanding of how rare POS tags can be better recognized with larger data sets.
The paper provides a comparison with several existing methods. However, it could be expanded. For example, it would be interesting to see a comparison of CRF with other state-of-the-art models on the same dataset, which would provide a more comprehensive assessment of the performance of the proposed approach. In addition, a discussion of the computational efficiency of the model compared to more complex models such as transformers could be included.
Expanding the discussion on error analysis could offer insights into why the model performed poorly on certain tags.

Reviewer 3 ·

Basic reporting

See attached PDF

Experimental design

See attached PDF

Validity of the findings

See attached PDF

Annotated reviews are not available for download in order to protect the identity of reviewers who chose to remain anonymous.

---

## Round 0.2 · accepted · Accept

Dear Authors,

I am grateful for your efforts in revising the paper. Reviewers have indicated that your revised paper is suitable for acceptance in its current form. I am also satisfied with the revised manuscript and believe it is now ready for publication.

Best wishes,

Reviewer 2 ·

Basic reporting

The paper is written in clear, professional English throughout. The language used is appropriate for an academic article.
The structure of the paper is clear and follows discipline norms.
The figures and tables provided are relevant and add value to the text.

Experimental design

The article's scope falls well within the aims of a journal focused on NLP and computational linguistics, particularly given its focus on POS tagging for a low-resource language like Urdu.
The methodology is clearly described, and sufficient detail is provided for replicating the experiments.

Validity of the findings

The findings are valid, well-supported by the experimental results, the accuracy achieved (96.1%) is impressive.
The experiments performed are thorough, with results demonstrating the effectiveness of the CRF-based model in comparison to existing methods. The authors provide adequate justification for their model’s superiority.

Additional comments

Overall, the authors took into account and responded to my comments, improving the clarity of the methodology, the description of how the model works, and its applicability in the future.

Reviewer 3 ·

Basic reporting

no comment

Experimental design

no comment

Validity of the findings

no comment

Additional comments

I accept the editing the authors made.